# Handheld Ultrasound or Conventional Ultrasound Devices in Patients Undergoing HCT: A Validation Study

**DOI:** 10.3390/jcm12020520

**Published:** 2023-01-08

**Authors:** Andrea Duminuco, Alessandra Cupri, Rosario Massimino, Salvatore Leotta, Giulio Antonio Milone, Bruno Garibaldi, Giulia Giuffrida, Orazio Garretto, Giuseppe Milone

**Affiliations:** 1Unità di Trapianto Emopoietico, Divisione di Ematologia, Azienda Ospedaliera Policlinico “G.Rodolico-San Marco”, 95123 Catania, Italy; 2Radiologia CAST Azienda Ospedaliera Policlinico “G.Rodolico-San Marco”, 95123 Catania, Italy; 3Istituto Oncologico del Mediterraneo, Unità di Trapianto Emopoietico, Divisione di Ematologia, 95029 Viagrande, Italy

**Keywords:** hematopoietic stem cell transplantation, handheld ultrasound device, liver veno-occlusive disease

## Abstract

Abdominal ultrasound exams play a major role in the diagnosis of sinusoidal obstruction syndrome/veno-occlusive disease (SOS/VOD). The development of portable hand-held ultrasound devices (HHUS) has been shown to facilitate the diagnosis of many diseases, but little data on the value of HHUS in the diagnosis of SOS/VOD are available. We performed a study aimed at validating portable ultrasound (US) devices in the setting of hematopoietic stem cell transplant (HCT). Sixteen evaluable patients undergoing allogeneic HCT were studied using conventional US and HHUS during the first 3 weeks after transplant. The results obtained demonstrate that there is a close correlation between conventional and handheld ultrasound examination in the measurement of the right hepatic lobe (r = 0.912, *p* < 0.0001), the left hepatic lobe (r = 0.843, *p* < 0.0001), the portal vein (PV) (r = 0.724, *p* < 0.0001), and the spleen (r = 0.983, *p* < 0.0001) based on Pearson’s correlation. The same data, analyzed through Lin’s concordance correlation coefficient, evidenced a substantial level of agreement in the comparison of the spleen and right hepatic lobe, while a lower grade of agreement in the measurement of the portal vein and left hepatic lobe. Moreover, there was good agreement between results obtained by the two types of ultrasound devices in assessing ascites (*p* < 0.0001), gallbladder thickening (*p* < 0.0001), and the direction of PV flow (*p* < 0.0001). HHUS device allows the study of HokUs-10 parameters with an excellent agreement with conventional US, and may contribute to SOS/VOD diagnosis.

## 1. Introduction

Hematopoietic stem cell transplantation (HCT) is a standard treatment for hematological diseases. One possible complication that may arise in the weeks following the transplant is sinusoidal-obstructive syndrome (SOS), also known as veno-occlusive disease (SOS/VOD) [1,2,3,4]. SOS/VOD could rapidly lead to patient death if not promptly diagnosed and treated [5,6]. Clinical criteria for diagnosis have been established [4,7,8,9]; however, their diagnostic sensitivity and specificity are limited [10], and improvement is needed in the identification of predictive factors for SOS/VOD. Ultrasound (US) plays a fundamental role in diagnosis [9,11,12]. Recently, a US scoring system denominated HokUS-10 has been reported to have high sensitivity and specificity when assessed at the time of clinically overt disease [13].

HHUS has gained acceptance in the diagnosis of many conditions in the last decade [14,15,16], but, in the context of VOD diagnosis, the place of handheld ultrasound devices (HHUS) is still undefined. Conventional and handheld ultrasound has been compared in various clinical settings with a good overall agreement between the two methods [17]. HHUS has been validated in the context of pleural diseases, in patients presenting abdominal emergencies, obstetrics and gynecologic affections, or vascular diseases [17]. HHUS may be particularly useful in the hematopoietic transplant ward, given its ease of usage and that it may allow for serial repetition of the evaluation. Further, if the ward has no on-floor access to conventional US, the use of HHUS avoids the need to transfer patients to the radiology department. The validation of HHUS, however, should be performed in each specific clinical setting in which it is to be used [17], and only scanty data on hand-held US in patients who underwent HCT are available [18,19].

Since a comprehensive comparison with the conventional US is lacking, we conducted a study to validate HHUS in the setting of the early phase of HCT.

## 2. Materials and Methods

### 2.1. Study Design

This prospective study aimed to compare a portable ultrasound instrument (handheld ultrasound) and validate it versus conventional ultrasound, in the analysis of the abdomen in patients receiving an allogeneic HCT. HHUS was operated by physicians from our BMT Unit, while the conventional US were operated by radiologists of the Department of Radiology. To avoid acquisition bias, the operator of conventional US and those who used HHUS were not allowed to know the results obtained by the counterpart. Informed consent was obtained from each patient. 

### 2.2. HHUS Device

We used the Butterfly iQ+™ ultrasound probe, powered by Ultrasound-on-Chip™ technology (2D array, 9000 micro-machined sensors). The probe can emulate any transducer-linear, curved, or phased- with the possibility of using different modes (M-mode, B-mode, Color Doppler, Power Doppler, Pulsed Wave Doppler). Specifically, the abdomen preset uses harmonic frequencies in a curvilinear format to support the clinical assessment of structures, the size and morphology of abdominal organs, such as the liver and kidney, to a depth of 25 cm. A USB-C cable connected this probe to an Apple iPhone™ (iPhone 11 or 13 Pro Max™).

### 2.3. Conventional Device

Conventional US exams were obtained using a General Electric Logiq E9™ ultrasound machine. This instrument uses advanced transducers with a single crystal, acoustic amplification, and cooling technology to improve image quality. The probe was a General Electric™ convex device with a 2–5 MHz transducer and a field of view of 69 degrees, indicated for abdomen study. 

### 2.4. Operators

The HH US examiners had previously achieved experience using ultrasound devices during a 6-month training period carried out by radiologists. The conventional ultrasound exams were operated by one of two different radiologists, each with more than 5 years of experience in ultrasound examination.

### 2.5. Time Points of Validation

Enrolled patients underwent ultrasound examination at specific time points during hospital admission, first with a handheld ultrasound device and subsequently within 4 h, with conventional ultrasound equipment.

Each ultrasound examination pair was performed at the following times:

- at day −10 (before starting conditioning regimen); at day +0 (day of the HSC infusion); at day +7; at day +14; and at day +21.

Ultrasound monitoring was performed regardless of the presence of changes and clinical signs (hyperbilirubinemia, abdominal pain, fluid retention, or other symptoms suggestive of SOS/VOD). Unplanned handheld ultrasound examination was also repeated in case of the appearance of suspicious signs or symptoms. Unplanned data were not used to validate the method.

### 2.6. Parameters Considered in the Ultrasound Examination

A common ultrasound scanning approach was established at the start of the study among operators, as already described (Figure 1) [20]. 

The ultrasound parameters sought with both radiological techniques are those described in the HokUS-10 scoring system for the diagnosis of SOS/VOD (Table 1). The results obtained with handheld ultrasound were expressed in terms of correlation indices between the two methods. Grade of steatosis was scored in four degrees from 0 to 4 as described [21]: absent (score 0); mild (score 1), if there is standard visualization of the diaphragm and the portal vein wall; moderate (score 2), if there is slightly impaired appearance of the portal vein wall and the diaphragm; severe (score 3), in case of a marked rise in liver echogenicity with poor or no visualization of structures.

The ultrasound images were stored digitally during the examination on both instruments. 

### 2.7. Patients 

We enrolled 17 patients undergoing allogeneic HCT in our center. The eligibility criteria for this study were adult age (>18 years), any underlying hematological disease, any stage of disease, full myeloablative or reduced-intensity radio-chemotherapy conditioning, and related or unrelated donors. The patient demographics and the transplant characteristics of the enrolled patients are shown in Table 2. 

The mean age of participants was 52.5 (range: 23–68) years. The most frequent hematological underlying disease was acute myeloid leukemia (47.1%), followed by myelodysplastic syndrome (17.7%), and primary myelofibrosis (17.7%). Other diagnoses accounted for 17.7%. 

Six patients (35.2%) received the transplant from an HLA-identical related donor, three patients (17.6%) from a haploidentical donor stem cell source, and eight patients (47.1%) from an unrelated donor. In two patients, the diagnosis of SOS/VOD was made based on the Seattle [8] or Baltimore criteria [4].

Of the seventeen patients initially enrolled, one patient did not undergo ultrasound measurements after he was diagnosed as infected with the COVID-19 virus early after transplant. He was placed in precautionary isolation limiting interactions with health personnel, and did not receive further US examinations.

### 2.8. Statistical Analysis

The degree of association of results obtained with the two different devices, if variables were categorical, were studied in a contingency table and analyzed by Chi-test or Fisher’s test: gallbladder wall thickening, amount of ascites, paraumbilical vein mean velocity ≤2 mm, direction of portal vein flow, appearance of paraumbilical blood flow, grade of steatosis. Cohen’s kappa statistic was used to evaluate the agreement between categorical results obtained with these two different devices. The agreement between HHUS and conventional devices in the detection of continuous variables hepatomegaly, spleen enlargement, and portal vein diameter, was compared through correlation analysis (Pearson’s correlation), with *p*-value < 0.05 considered significant, and Lin’s concordance correlation coefficient (if >0.95, it was considered substantial). In addition, Bland–Altman’s plots related to each individual patient were performed. The diagnostic performance of HHUS was measured in terms of sensitivity and specificity. To this aim, the results of the conventional ultrasound were assumed as the gold standard. Statistical analysis was performed using R-commander software (R Foundation for Statistical Computing, Vienna, Austria. https://www.R-project.org/). Final data analysis was performed on 08, 2022.

## 3. Results

### 3.1. Clinical and Laboratory Features 

Of the 17 patients enrolled in the study, 16 (94.2%) received ultrasound examinations at the established time points, and 1 (5.8%) did not, since after the infusion of stem cells he was diagnosed with COVID-19. He was subsequently cared for using strict isolation measures, and no data obtained from this patient were considered. At 60 days after transplantation, 17/17 (100%) were alive. On day +60, 16/17 achieved complete remission of the hematological disease with full-donor chimerism (>95%), and 1 (5.9%) experienced an early relapse of AML.

Twelve out of seventeen patients presented fever (70.5%), and bacteremia was diagnosed in five patients (29.4%). Nine out of seventeen patients (52.9%) experienced acute GvHD grade II-IV. All of them were treated with first-line corticosteroid therapy, and cortico-refractory-GVHD was treated with extracorporeal photopheresis or ruxolitinib. SOS/VOD was diagnosed in two patients, in one patient (5.8%) based on Baltimore criteria (6,7), and according to Seattle criteria in the second. At day +6, the first patient experienced an increase in bilirubin value (maximum 4.41 mg/dL), weight gain (an increase of 10% from baseline), and the presence of ascites and liver enlargement. Therapy with defibrotide at a standard dosage was begun promptly, with complete resolution. The second patient meeting the modified Seattle criteria, on day +7, had ascites, hepatomegaly, body weight gain (8%), and an increase in portal vein diameter without a concomitant rise in bilirubin values (0.72 mg/dL). His clinical condition, however, returned to normal with supportive therapy only. Conventional US and HHUS detected the same radiologic signs in both patients. 

### 3.2. Comparison between Conventional and Handheld Ultrasound Measurement

A total of 80 abdomen ultrasound paired exams (80 with the conventional device and 80 with the Butterfly iQ+™ ultrasound probe) were performed and compared (5 pairs of measurements for 16 patients). 

The correlations between the individual measurements of the criteria, listed among the HokUS-10 Parameters, were evaluated. There was a correlation between conventional and handheld ultrasound examination in the measurement of the right hepatic lobe (Figure 2A, Pearson correlation = 0.912, *p* < 0.0001) or the left hepatic lobe (Figure 2B, Pearson correlation = 0.843, *p* < 0.0001), of the diameter of the portal vein (Figure 2C, Pearson correlation = 0.724, *p* < 0.0001), and of the dimension of the spleen, (Figure 2D, Pearson correlation = 0.983, *p* < 0.0001). The same data evaluated, according to the concordance correlation coefficient (ρ_c_), demonstrated a substantial correlation for the measurement of the spleen (ρ_c_ = 0.974) and right hepatic lobe (ρ_c_ = 0.991). Conversely, the agreement between the left hepatic lobe and the portal vein was lower (respectively, ρ_c_ = 0.834, and ρ_c_ = 0.713).

The Bland–Altman’s plots related to each individual patient are reported in the Appendix A. For each patient, the degree of agreement according to Lin’s concordance correlation coefficient was variable, between substantial (ρ_c_ > 0.95 to 0.99, 12/16 patients) and near perfection (ρ_c_ > 0.99, 4/16).

Significant associations were found among the two techniques on the detection of ascites (*p* = 0.0001), gallbladder wall thickening >6 mm (*p* = 0.0009), the direction of PV flow (*p* = 0.0001), and appearance of PUV blood flow signal (*p* = 0.0001) (Table 3). 

Concordant results among the two techniques were also obtained in the assessment of the grade of steatosis (Chi-square *p*-value ≤ 0.0001) (Figure 3). 

Raw data are reported in a Appendix A.

HHUS correctly recognized right lobe hepatomegaly and left lobe hepatomegaly, with a sensitivity of 95% and 100% and a specificity of 87% and 50%. Gallbladder wall thickening measurement was correctly recognized with 100% sensitivity and 98% specificity. 

### 3.3. HHUS in the Diagnosis of Other Radiologic Abnormalities

Other lesions encountered were biliary cysts (in two cases), liver fungal hepatic lesions (in forty-four cases), and calcific micronodules in the spleen (three cases). In all instances, these lesions were detected both using conventional and handheld techniques.

## 4. Discussion

Our study was aimed at evaluating the reliability and applicability of HHUS after bone marrow transplantation. The choice of the HHUS methodology was based mainly on the ease of repeating the assessment allowed by HHUS. We think, in fact, that in the transplant patient, this feature may make HHUS of greater utility than a conventional ultrasound.

The diagnosis of SOS/VOD is difficult and controversial. The clinical criteria that are the basis for the diagnosis (namely modified Seattle criteria, Baltimore criteria, EBMT criteria [4,8,9]) have limited sensitivity and specificity. In patients suffering from VOD as diagnosed by the Seattle criteria, the repeated assessment with the application of the Baltimore criteria resulted in a substantial proportion of false negatives, i.e., patients who showed a positive biopsy and positivity to the Seattle criteria, but negativity to the Baltimore criteria so that the sensitivity of the Baltimore criteria was only 56% [10]. Additionally, in other studies, the sensitivity of the Baltimore criteria appears low relative to the Seattle criteria; Corbacioglu [22] reported that of 780 patients who meet the Seattle criteria, 331 do not meet the Baltimore criteria. Carreras et al. reported that over 117 patients diagnosed as affected by SOS/VOD, according to the Seattle criteria [23], 37.6% (*n* = 44) did not develop hyperbilirubinemia, and therefore did not fulfill the Baltimore criteria. The Seattle criteria, although proving greater sensitivity, lacks, on the other hand, specificity. In the previously mentioned set of 30 biopsies, in patients with VOD based on the Seattle criteria, it was found that 18/30 (60%) indeed had histological signs of SOS/VOD, but the diagnosis of VOD was not confirmed on histology in the remaining 12 patients (40%).

Furthermore, a considerable number of patients could remain undiagnosed with the application of current criteria, and even evolve toward TRM. Recently, the presence of SOS/VOD that remained undiagnosed was found in about 20% of patients who developed multi-organ dysfunction [24].

Further, the difficulty of diagnosis is also linked to the fact that SOS/VOD requires the exclusion of alternative causes in each case. In the study of Carreras based on biopsy, out of twelve patients with suspected VOD according to the Seattle criteria, five (15%) were diagnosed with hepatic GVHD at histology [10]. However, some of these etiologies, such as drug-induced liver disease, are difficult to deal with.

The recently proposed EBMT criteria [9] do not substantially modify these limitations, as they substantially repropose the diagnostic criteria of Baltimore with only minor modifications.

Abdomen Ultrasound examination plays a major role in the diagnosis of SOS/VOD, [11]. Ultrasound is strongly recommended by the BSH and EBMT guidelines and constitutes a cornerstone of diagnosis in pediatric age, as the evaluation of hepatomegaly and ascites through imaging is required in the EBMT criteria for the diagnosis of VOD in children. Ultrasound signs include hepato-splenomegaly, ascites, periportal and gallbladder wall edema, increased width of the portal vein, indistinct borders or narrowing of hepatic veins. Doppler-mode characteristics include portal vein flow demodulation, decreased flow velocity or reversal of portal vein flow, decreased spectral density, increased resistive index and peak systolic velocity of the hepatic artery, congestion index, monophasic reduced flow in the hepatic veins, and visualization of collateral veins (e.g., in the paraumbilical veins). However, the most common ultrasound signs are nonspecific [25].

As has been recently pointed out [26], the main limitation of published ultrasound studies is the fact that they lack serial repetition during the transplant. Most of these studies have, indeed, evaluated the diagnostic capability of the ultrasound, with data only obtained when the clinical diagnosis is reached [27].

In the same way, as in the Nishida study [13], the predictive and diagnostic value of the HokUS-10 ultrasound score is determined on ultrasound scans carried out mainly on day +14, a time at which a substantial number of SOS/VODs have already had a clinical diagnosis. Data collected in this way are of limited use, as to have the greatest chance of therapeutic success, it is important to obtain the diagnosis as early as possible [5,6]. Diagnostic assessments by other instruments, such as CT-scan and MRI, are not easily repeated as they implicate the need for patient transportation and expose the patient to risks of toxicity, so they are indicated only in the baseline assessment before transplant [20]. Elastography is a relatively new technique and has promising results [20]; its repeatability is, however, problematic for logistical reasons.

Instead, it would be important to carry out more frequent ultrasound scans, beginning during conditioning, with a frequency even higher than weekly. The use of the HHUS can overcome the main weakness of a conventional ultrasound, and of other diagnostic instruments, which is poor repeatability.

HHUS had not been validated in the transplant field, therefore we considered it appropriate to validate it in the diagnosis of the US signs recognized as important (HOKUS-10). The data from the validation study collected by us, with conventional ultrasound and portable ultrasound, show that HHUS is reliable in detecting the ultrasound abnormalities proposed in the HokUS-10 scheme for SOS/VOD. Concerning the lower agreement through Lin’s concordance correlation coefficient reached by the left hepatic lobe and portal vein, the explanation could lie in the fact that the measurement of these parameters is extremely subjective and even the smallest difference in the execution of the exam could give important differences between the two methods. In any case, as visible in the individual Bland–Altman’s plots, each method was able to identify the alterations suggestive of SOS/VOD. Our validation study was planned and conducted prospectively, even though the number of patients studied was limited. We underline that the study preparation took some months, which was necessary to obtain the dedicated probes and to train the staff.

## 5. Conclusions

In conclusion, HHUS in the transplant setting allows for reliable and serial data collection. In this way, HHUS could substantially contribute to the identification of predictive anomalies of SOS/VOD, the reduction of underdiagnosed VOD, and to the differential diagnosis of liver changes occurring during HCT.

## Figures and Tables

**Figure 1 jcm-12-00520-f001:**
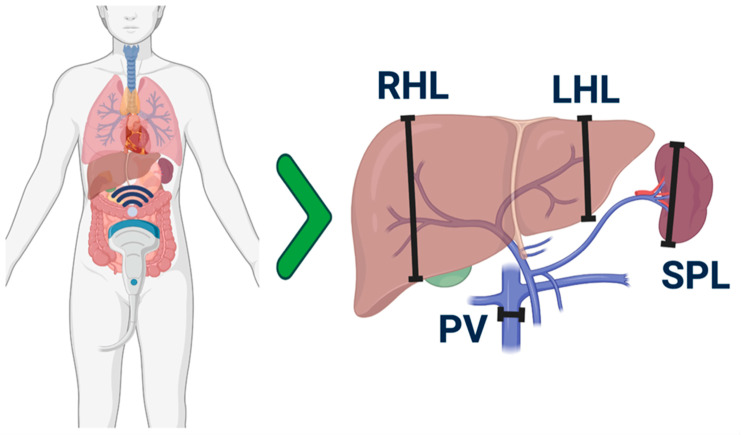
Schematic representation of anatomic sites where measurements were performed by US. The right hepatic lobe (RHL) was measured on the hemiclavicular right line and the left hepatic lobe (LHL) on the parasternal right line. Other measurements were concerning the spleen (SPL) and portal vein (PV).

**Figure 2 jcm-12-00520-f002:**
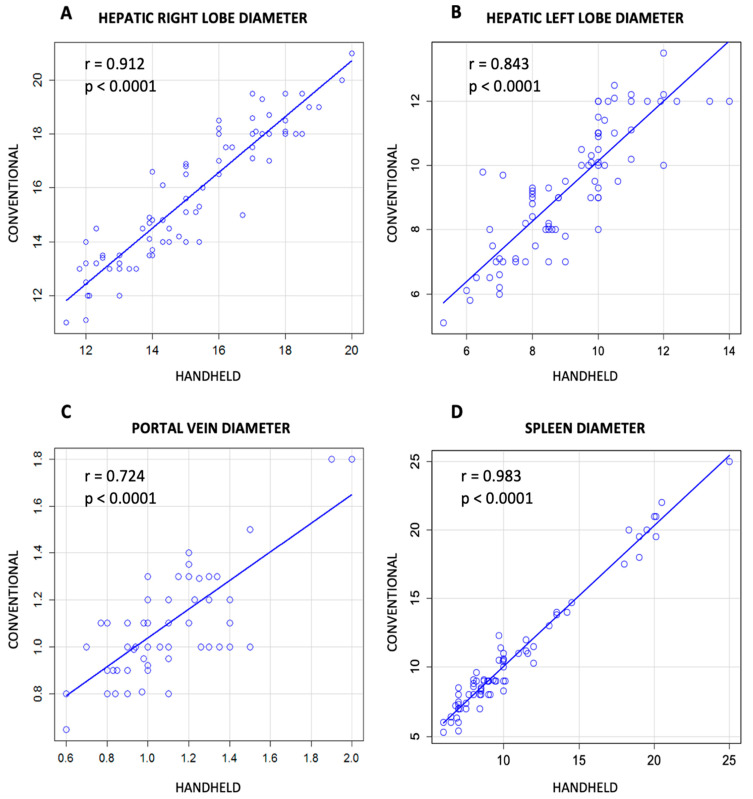
Correlation between results obtained with handheld and conventional ultrasound in measurement of diameter of liver ((**A**) and (**B**) for right and left hepatic lobe diameter, respectively), portal vein (**C**), and spleen (**D**).

**Figure 3 jcm-12-00520-f003:**
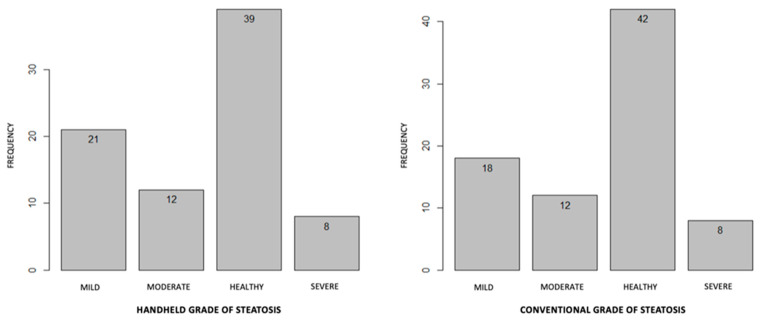
Frequency of degrees of hepatic steatosis with conventional and HHUS methods. Chi-square, *p* < 0.0001.

**Table 1 jcm-12-00520-t001:** List of the Parameters required by HokUS-10 US score system to diagnose SOS/VOD and additional parameters analyzed.

HokUS-10 Parameters	Definition Threshold
Hepatic left lobe vertical diameter	≥70 mm
Hepatic right lobe vertical diameter	≥110 mm
Gallbladder wall thickening	≥6 mm
PV diameter	≥12 mm
PUV diameter	≥2 mm
Amount of ascites	Moderate to severe
PV mean velocity	≤2 mm
Direction of PV flow	Congestion or hepatofugal
Appearance of PUV blood flow signal	Yes
Hepatic artery Resistive Index	≥7.5
Further parameters analyzed in this study	
Spleen size	
Hepatic lesions	
Spleens lesions	
Steatosis’ grade	

PV: portal vein, PUV: paraumbilical veins.

**Table 2 jcm-12-00520-t002:** Baseline and transplant characteristics of patients enrolled.

Age (Years)	Mean (Range)	52.5 (23–68)
Sex, males/females	N (%)	7 (41.2)/10 (58.8)
Hematological disease - AML - HL - MDS - PMF - ALL	N (%)	8 (47.1) 2 (11.7) 3 (17.7) 3 (17.7) 1 (5.8)
Refined DRI - Low - Intermediate - High	N (%)	10 (58.8) 4 (23.5) 3 (17.7)
Status disease - Complete response/remission - Partial response/remission - Active disease	N (%)	12 (70.6) 2 (11.7) 3 (17.7)
Transplant type - Related - Unrelated	N (%)	9 (52.9) 8 (47.1)
Cells source - PBSC - Bone marrow	N (%)	14 (82.3) 3 (17.7)
HLA match - 7/8 - Full matched - Haploidentical	N (%)	1 (5.8) 13 (76.5) 3 (17.7)
Conditioning regimen - Myeloablative - Non myeloablative	N (%)	17 (100) 0 (0)
HCT-CI - 0 - 1	N (%)	14 (82.3) 3 (17.7)
ATG use, yes/no	N (%)	6 (35.3) /11 (64.7)
Karnofsky Performance Score - ≥90 - <90	N (%)	17 (100) 0 (0)
Basal Bilirubin (mg/dL)	Mean (range)	0.56 (0.3–1.15)
Basal Creatinine (mg/dL)	Mean (range)	0.72 (0.32–1.18)

AML: acute myeloid leukemia; HL: Hodgkin lymphoma; MDS: myelodysplastic syndrome; PMF: primary myelofibrosis; ALL: acute lymphoblastic leukemia; DRI: disease risk index; PBSC: peripheral blood stem cells; HLA: human leukocyte antigen; HCT-CI: HCT-Comorbidity Index; ATG: anti-thymocyte globulins.

**Table 3 jcm-12-00520-t003:** Results obtained by conventional and handheld ultrasound on assessment of ascites, gallbladder wall thickening, and para-umbilical veins.

HokUS-10 Parameters	Findings-Conventional US/Handheld US				*p*-Value	Kappa Statistic for Agreement between the Two Tests
	Yes/Yes	Yes/No	No/Yes	No/No		
Ascites	7	1	1	71	<0.0001	0.93
Gallbladder wall thickening	2	0	1	77	<0.0009	0.79
Appearance of PUV blood flow signal	0	0	0	80	<0.0001	1.0
	Hepatopetus conventional US/ handheld US		Hepatofuge conventional US/ handheld US			
Direction of PV flow	80/80		0/0		<0.0001	1.0

## Data Availability

Data are stored in a data worksheet and can be made available upon request.

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
