# Peer review of "Handheld Ultrasound or Conventional Ultrasound Devices in Patients Undergoing HCT: A Validation Study"

_jcm, 2023, doi:10.3390/jcm12020520_

Round 1

Reviewer 1 Report

The manuscript is well written. However, there is a major issue on the statistics used. Is there a statistician among the authors?

Major:The authors didn't performed statistics adequately. They consider the 80 pairs of measures independent while there are performed on 17 patients. The factor "patients" have to be taken into account in such a case. The data has a tridimensional unit level: first the patient, second the localisation of the measure and third the day of measurement. These three levels have to be taken into in a model.

Minor: In the statical part, I discourage the use of "We".

Author Response

reply:

The advice of a statistic expert was asked ( Dr Giovanni Tripepi - Reggio Calabria) and we modified the analysis according to his advice.

16 patients were studied in all. All were studied at 5-time points (16x5= 80 evaluations). Thus, each patient was studied the same number of times.

We completed the comparison of the two ultrasound methods using also "Lin's concordance correlation coefficient" (as declared in the abstract, in the methods and in the results sections)

In each patient, we also obtained a Bland-Altman’s plots and all such plots are now supplementary files.

Reviewer 2 Report

This is a well conducted prospective study comparing portable HHUS with radiology department US in HCT patients.

The paper is very well written with all important references regarding SOS/VOD in HCT are included.

Minor:HSCT should be changed to HCT(Hematopoietic cell transplantation).

Author Response

HSCT has benn changed to HCT(Hematopoietic cell transplantation).

Round 2

Reviewer 1 Report

The authors answered adequately to my comments.

The results are now adequately presented.

Typos:

In the abstract it is mentioned 17 patients while in the authors replies, 16. This must be fixed. I think the difference comes from the fact that one patient was COVID positive and not evaluated for HHUS.

Table 2: The number and percent for AML are in italic.

Author Response

in the abstract now we declare 16 patients

italic in the table has been corrected.